# Preschool Children’s Eating Habits and Parental Nutritional Status

**DOI:** 10.3390/nu17030575

**Published:** 2025-02-05

**Authors:** Silvia Bettocchi, Veronica D’Oria, Valentina De Cosmi, Silvia Scaglioni, Carlo Agostoni, Luigi Paglia, Michela Paglia, Sara Colombo, Francesca Braiotta, Matteo Beretta, Cristiana Berti

**Affiliations:** 1Pediatric Unit, Fondazione IRCCS Ca’ Granda Ospedale Maggiore Policlinico, 20122 Milan, Italy; pilla.sma@gmail.com (S.B.); cristiana.berti@policlinico.mi.it (C.B.); 2Pediatric Intensive Care Unit, Fondazione IRCCS Ca’ Granda Ospedale Maggiore Policlinico, 20122 Milan, Italy; veronica.doria@policlinico.mi.it; 3Department of Food Safety, Nutrition and Veterinary Public Health, Istituto Superiore di Sanità—Italian National Institute of Health, 00161 Rome, Italy; valentina.decosmi@iss.it; 4Fondazione De Marchi, Fondazione IRCCS Ca’ Granda Ospedale Maggiore Policlinico, 20122 Milan, Italy; silvia.scaglioni@unimi.it; 5Department of Clinical Sciences and Community Health, Università degli Studi di Milano, 20122 Milan, Italy; 6Department of Maternal and Child Dentistry, Istituto Stomatologico Italiano (ISI), 20122 Milan, Italy; luigipaglia@hotmail.com (L.P.); michelapaglia@hotmail.it (M.P.); sara.colombo.mi@gmail.com (S.C.); fbraiotta@hotmail.it (F.B.); 7Odontoiatria Materno Infantile (OMI Benefit), 20123 Milan, Italy; teoberet@libero.it; 8Orthodontic School, Università degli Studi di Brescia, 25121 Brescia, Italy

**Keywords:** children feeding habits, parental overweight/obesity, sugar-sweetened beverages, breastfeeding, dietary intake

## Abstract

**Background/Objectives:** Poor nutrition in early life represents a relevant public health issue globally. The current study aimed to characterize eating habits among preschoolers and investigate the relationship with parents’ nutritional status. **Methods:** A secondary analysis of a cross-sectional study carried out at the Child Dentistry Clinics of the Istituto Stomatologico Italiano, Milan, Italy, including 171 patients aged 12–71 months and their parents, was conducted. Demographic data, anthropometric measurements, and information on children’s eating habits were collected. **Results:** The duration of exclusive breastfeeding was 5.9 (±6.8) months in 65% of children, and complementary feeding began at a mean (±SD) of 6.2 (±1.7) months in accordance with present recommendations. In contrast, the consumption of fruit juice 3.2 (±2.9) times/week, a protein intake of 3.0 (±0.6) g/kg, and a sugar intake of 20 (±8)% were over guideline limits. Overweight/obese children were introduced to cow’s milk earlier (*p* = 0.033) and consumed a higher percentage of total fats (*p* = 0.026) whilst consuming a lower percentage of carbohydrates (*p* = 0.050). In terms of children with both parents being obese or overweight, they consumed more carbohydrates (*p* = 0.048). Finally, we found that birth weight correlated positively with maternal BMI (ρ = 0.159; *p* < 0.05). The number of offspring correlated positively with the weekly frequency of sugar-sweetened beverage consumption before 24 months (ρ = 0.282; *p* < 0.001) whilst correlating negatively with the age of cow’s milk introduction (ρ = −0.226; *p* < 0.01). **Conclusions:** Our findings, according to recommendations, suggest that the prevention of obesity needs to begin in infancy. As parents play a pivotal role in establishing children’s food choices, nutritional education aimed at families is needed.

## 1. Introduction

Nutrition in the first years of life may impact an individual’s health, well-being, and potential, with lifelong consequences. Exclusive breastfeeding (EB) is recommended for the first 6 months to achieve optimal growth, development, and health, with the introduction of appropriate complementary foods thereafter and continued breastfeeding for up to 2 years or beyond [1]. During this period, the basis for future eating patterns is established, which may influence the global increase in pediatric overweight and obesity and consequently in Non-Communicable Diseases (NCDs) and early childhood caries. Furthermore, neophobia, food refusal, or selectivity during mealtime are common behaviors among toddlers and children under the age of 5, leading to important limitations in the variety and number of foods consumed, which may compromise growth and the development of healthy eating habits [2]. Infants and young children learn about what, when, and how much to eat through both the experience of eating and the family context around food [3]. Parents and guardians influence them both directly and indirectly, adopting overt and covert control, and they play a direct role in shaping their child’s eating behavior by deciding how to feed them or how to deal with eating problems and by serving as models for dietary choices and patterns [4]. Moreover, the parental Body Mass Index (BMI) category is likely a predictor of a child’s eating behavior and BMI [5]. In particular, a study including English and Welsh families provides evidence that, in comparison to children belonging to families with normal-weight parents, those from obese/overweight families demonstrate a higher preference for fatty foods, a lower liking for vegetables, and a more “overeating-type” eating style [6]. Similarly, girls from families with one or both parents being overweight, living in Pennsylvania, snack more frequently with higher intakes of fat from energy-dense snacks [7]. Furthermore, a study conducted in South Carolina shows that families with an underweight or healthy-weight parent reported healthier habits, that is, a less obesogenic home environment, compared to families with an overweight or obese parent [8]. In particular, data about families from Burlington or Akwesasne (USA) suggest that maternal obesity tends to influence dietary fat intake in children [9].

To date, there is a shortage of data about childhood eating behaviors in Italy and their relationship with parents’ nutritional status, especially in the first years of life. Therefore, the present study aimed to (1) characterize feeding practices, eating behaviors, and nutrient intakes among preschool-aged children; and (2) investigate the relationship between children’s eating and their parents’ nutritional status.

## 2. Materials and Methods

This study represents a secondary analysis using data from a one-year cross-sectional study including patients aged 12–71 months and their parents. Study details were published elsewhere [10]. Figure 1 depicts the study design. Written informed consent was obtained from the parents (or legal caregiver) of each patient.

### 2.1. Demographic Data

A survey was performed by trained personnel to collect both parents’ socio-demographic and smoking information and their child’s health history.

### 2.2. Anthropometric Measurements

Parents’ reported weights and heights were gathered, and BMI was calculated. Child body weight was measured using a gram scale (Tanita TL-150 MA; Sensor Medics, Milano, Italy), accurate to 0.1 kg; child length (12–23 months) was measured with an infantometer (SECA 416, Hammer Steindamm 3-25 22089, Hamburg, Germany); and height was measured with a stadiometer (SECA 213, Hammer Steindamm 3-25 22089, Hamburg, Germany). BMI was calculated as weight (kg)/length or height (m^2^). *Z*-scores and percentiles for weight for age and weight for length were calculated using World Health Organization (WHO) Anthro and Anthro Plus^®^ software (http://www.who.int/growthref/tools/en/, accessed on 30 June 2023) and WHO reference charts [11]. Nutritional status was classified according to WHO criteria [12].

### 2.3. Children’s Eating Behavior Inventory (CEBI)

Child eating and mealtime problems were assessed using a specific 40-item parent-report questionnaire, i.e., the CEBI, investigating children’s food preferences, behavioral compliance, and motor-based feeding skills [13]. The instrument evaluated parents’ feelings about feeding their child and interactions between family members too. An Eating Behavior score (EBs) of 92 plus a Parent-Perceived Eating Problems (PPEP) score of >16% identified children at risk for eating problems [14].

### 2.4. Children’s Eating Habits

Early-life habits were investigated through an ad hoc questionnaire to record the following: breastfeeding duration and classification according to WHO definitions, formula and cow’s milk introduction, night feeding, complementary feeding, and sweet beverage introduction. A validated age-adjusted Food Frequency Questionnaire (FFQ) consisting of 99 items was administered by a trained registered dietitian nutritionist to the parents to assess the most recent dietary habits of their child [14]. The quantification and analysis of energy intake and nutrient composition were performed by Metadieta^®^ PC software program 4.7 (METEDA S.r.l, Via Antonio Bosio, 2–00161 Roma (RM)). The daily amounts of energy, proteins, total fats, carbohydrates, sugars, fibers, iron, and calcium per day were recorded.

### 2.5. Statistical Analysis

A sample size of 157 subjects was necessary to achieve a precision of 5% with a 95% significance level [10].

Categorical or ordinal variables were expressed as the frequency (percentage, %); continuous variables were expressed as the mean and standard deviation if normally distributed, and as the median and interquartile range if not. Within-group and between-group comparisons were performed with parametric or non-parametric statistical tests, where appropriate: for comparisons between groups, Student’s *t*-test (2 groups) or ANOVA (>2 groups) was typically used for distributed continuous variables, the Mann–Whitney test (2 groups) or Kruskal–Wallis test (>2 groups) for asymmetrically distributed continuous variables, and the Chi-square test or Fisher’s exact test (where appropriate) for categorical variables. To measure the possible two-way linear association between two variables, Pearson’s and Spearman’s correlation coefficients were used when appropriate.

The level of statistical significance was set at a *p*-value < 0.05, and, where appropriate, 95% CIs were calculated.

## 3. Results

### 3.1. Children

We reached a total of 176 patients. Five subjects refused to participate; thus, we enrolled 171 participants. Their mean age (±standard deviation) was 4.6 (±1.1) years (range 12–71 months), and 58% were males. Regarding their nutritional status, 78% were adequately nourished, 14% were underweight, 7% were overweight, and 1% were obese (Table 1). Considering the 24 underweight children, we found that one of them was also a low birth weight (LBW) newborn and two were both preterm and LBW.

Table 2 reports children’s feeding practices during infancy and at the time of the survey. The mean duration of EB was 5.9 (±6.8) months, and 111 children (65%) were exclusively breastfed for at least a period of 6 months. Complementary feeding began at 6.2 (±1.7) months, self-eating was achieved at 16.5 (±8.6) months, and children started consuming family foods at 17.8 (±8.9) months. Our population ate 4.8 (±1.6) meals/day.

When food and beverage preferences were investigated, pasta was shown to be the most preferred food item (78 subjects, 46%), followed by sweets and sweet bakery items (24 children, 14% with 11, 6% with preferences for chocolate). As for the most refused food items, vegetables were the least preferred (90 subjects 53%), with cruciferous vegetables being the most refused ones (16 subjects, 9%), followed by fish (18 subjects, 11%), whilst only 27 children (16%) were declared to refuse no food. Among beverages, water was the most preferred (63 subjects, 37%), followed by cola-based beverages (36 children, 21%), tea-based drinks (28 subjects, 16%), and fruit juices (25 children, 15%). The consumed sugar-sweetened beverages (SSBs) consisted mostly of fruit juice, whilst chamomile, tea, or herbal tea were less consumed. Fruit juice was introduced at a mean age of 2.2 (±1.3) years (with a minimum age of 6 months and 32 children (19%) consuming it by 1 year of age) and was drunk with a frequency of almost three times/week. Concerning chamomile, tea, or herbal tea, the minimum age at introduction was 4 months. We found that only 22 (13%) children never drank any of the SSBs.

Table 3 shows the current energy and nutrient intakes collected from 167 FFQs (4 participants did not complete the questionnaire).

The analysis of the CEBI indicated that 18 (11%) subjects had a PPEP score ≥ 16% and 142 (83%) an EBs ≥ 92, resulting in 17 children (10%) being at risk of eating problems. When this group was investigated in detail, we found that 11 (65%) children had at least one overweight/obese parent, whilst only 6 (35%) had healthy-weight parents.

Compared to not obese/overweight children (n = 158), overweight/obese children (n = 13) had significantly higher anthropometric variables (*p* < 0.05), were introduced to cow’s milk earlier [(12.4 ± 2.0) months versus (16.8 ± 9.4) months; *p* = 0.033], and consumed a higher percentage of total fat intake [(34.9 ± 9.3)% versus (29.2 ± 6.4)%; *p* = 0.026] whilst consuming a lower percentage of carbohydrates [(52.0 ± 5.3)% versus (54.9 ± 7.3)%; *p* = 0.050].

### 3.2. Parents

Table 4 shows parents’ characteristics. On average, mothers were 38.2 (±5.8) years old and fathers were 41.3 (±7.2) years old. Most parents worked, and the mothers reported the highest school level. At least a fourth of parents were smoking at the time of the survey.

Regarding nutritional status, 65% of the mothers and 48% of the fathers were of a healthy weight, 13% and 39% were overweight, and 8% and 12% were obese, while 10% and 1% were underweight (Table 5). When considering the distribution of parental nutritional status within families, we found that 71 (42%) children had one overweight/obese parent and 29 (17%) had both overweight/obese parents.

When considering children with overweight/obese parents (i.e., at least one or both) and those with no obese/overweight parents (i.e., underweight or healthy weight) (Table 6), we found that the former showed a lower percentage of total protein intake [(15 versus 16) %; *p* = 0.010] and a later introduction of growing-up milk [(14.9 versus 11.6) months; *p* = 0.032].

Moreover, both parents had a higher BMI (*p* < 0.01), whilst the mothers were younger [(37.3 versus 39.5) years; *p* = 0.013] and smoked more cigarettes [(2.5 versus 0.9); *p* = 0.013].

When focusing on both parents being overweight or obese (Table 7), we found that their children consumed more carbohydrates [(212 versus 199) g; *p* = 0.048] and were born at a higher weight [(3444 ± 462) g versus (3247 ± 506) g; *p* = 0.039;] with respect to children from no obese/overweight parents or with only one obese/overweight parent.

### 3.3. Correlations

We found positive correlations between the duration of exclusive breastfeeding and the age at the introduction of infant formula (ρ = 0.699; *p* < 0.001); the age at the introduction of growing-up milk and that of fruit juice (ρ = 0.484; *p* < 0.001); and the age at the introduction of chamomile and that of herbal tea (ρ = 0.927; *p* < 0.001). Maternal BMI positively correlated with child BMI (ρ = 0.170; *p* < 0.05), paternal BMI (ρ = 0.297; *p* < 0.001), and the offspring’s weight at birth (ρ = 0.159; *p* < 0.05). Furthermore, there was a positive correlation between child BMI and the weight at birth (ρ = 0.222; *p* < 0.01). Finally, we found that the number of offspring correlated positively with the weekly frequency of SSB consumption before 24 months (ρ = 0.282; *p* < 0.001) whilst correlating negatively with the age at the introduction of cow’s milk (ρ = −0.226; *p* < 0.01).

## 4. Discussion

Early-life nutrition, including breastfeeding duration, the time of complementary feeding, and food nutritional quality, may have several health implications in the short and long term (e.g., risk of developing obesity), with dietary habits shaped at a young age persisting over time. Parents play an active role in establishing eating behaviors that will continue throughout their child’s life.

In our population, the length of EB and the age at the introduction of complementary feeding were in line with recommendations [1]. Sixty-five percent of the children were exclusively breastfed for at least 6 months, indicating that, with respect to other Italian studies [15,16,17,18], in our population, mothers complied more properly with WHO recommendations. The breastfeeding practice has several health benefits, including a reduction in the odds of becoming obese among children breastfed for at least 6 months, as confirmed by the WHO European Childhood Obesity Surveillance Initiative study [19]. The age at the introduction of complementary food was approximately 6 months, as recommended to satisfy the child’s growing nutrient needs and support their healthy growth and development [1]. In this regard, it is worth recalling the importance of consuming a diverse diet, prioritizing high-nutrient-density foods (e.g., animal-source foods, fruits and vegetables, and nuts, pulses, and seeds) whilst minimizing processed or ultra-processed ones that are high in energy and low in nutrients. The onset of self-feeding occurred around 17 months, typically reflecting the toddlers’ more developed eating skills [20]. Contrary to recommendations, we found that a fifth of children consumed fruit juice by 1 year of age, and the overall consumption of fruit juice by our population was nearly three times/week. Likewise, a study conducted in five European countries showed that energy-providing liquids other than breast milk or formula were fed early (i.e., 4 months) and at a high rate to infants during the first year of life [21]. A similar trend was observed in a cross-sectional study involving 1- to 3-year-old children from ten Latin American countries [22]. Based on Italian recommendations from scientific bodies, sugary drinks should be avoided in the first 2 years of life [23]. The American Academy of Pediatrics recommends not introducing fruit juice to infants before 1 year of age, while allowing the consumption of 120 mL (nearly a glass) daily of fruit juice among toddlers, but only as part of a meal or snack [24]. The high consumption of fruit juice by our participants may partially explain their sugar intake being higher than the recommended value (~20% versus <15%) as suggested in the National Guidelines for Prevention [25]. Furthermore, we found that their protein intake (~3 g/kg) was almost three times higher than the reference value (~1 g/kg) [25]. There is evidence that a higher consumption of SSBs, that is, sugars, in the pediatric population increases the risk of overweight/obesity [26], dental caries [27], and NCDs [28]. Indeed, findings indicate a negative effect of excessive free sugar consumption on health and well-being for the human population as a whole [29]. Moreover, a daily intake of sugar-containing beverages in early childhood (≤2.5 years) was seen to be strongly associated with a greater intake in later childhood [30]. The first 2 years of life are crucial for the young child to establish lifelong flavor and food preferences. In terms of the innate preference for a sweet taste that infants are born with, repeated exposure to free sugars early in life may strengthen the desire for the sweet taste, resulting in a preference for and the consumption of sweet-tasting beverages and foods thereafter [31]. Instead, children need to experience a variety of textures, bitter and sour flavors, and nonsweet foods to learn to accept healthy foods and beverages. Overall, evidence suggests that sugary drinks should be avoided because they contribute calories with few nutrients whilst decreasing the child’s appetite for more nutritious foods [32]. Likewise, an excessive protein intake is likely associated with a later risk of being overweight or obese [33]. When eating habits were investigated based on the nutritional status of our children, we found that obese/overweight children were introduced to cow’s milk earlier than non-obese ones. Compared to breast milk, cow’s milk has more protein per unit energy content [34] at the expense of fats. Overall, a reduced energy balance owing to high-protein low-fat intake in early life followed by increasing fat intake with age is associated with a higher risk of later overweight and obesity [35]. In contrast, the protective effect of human milk against an early adiposity rebound likely depends on its low protein and high fat content, suggesting that the restriction of dietary fat in early life could increase susceptibility to being overweight. Moreover, obese/overweight children in our population had a higher percentage of fat intake.

When differences in child feeding practices were explored based on parental nutritional status, we found that more than half of children had at least one or both parents being overweight or obese, which indicates a potential risk factor for the development of obesity at any stage of life. Furthermore, despite the fact that we detected a low prevalence of children (10%) at risk of eating problems, two-thirds of them had at least one overweight/obese parent. A meta-analysis of studies across the world indicated that children of overweight or obese parents were >2 times more likely to be overweight or obese than those with healthy-weight parents [36]. The genetic and environmental factors (i.e., eating and lifestyle behaviors) shared between children and parents likely represent a familial predisposition to obesity, with there being concordance between parents and children in terms of weight trajectories [37] and dietary intake [38]. Consistent with the literature, we found that the children of both overweight or obese parents had a higher weight at birth with respect to those with no obese/overweight parents or only one obese/overweight parent, and maternal BMI correlated positively with the children’s birth weight and their current BMI. Moreover, compared to children with no obese/overweight parents or only one obese/overweight parent, those with both parents being overweight or obese had a higher intake of carbohydrates. Instead, contrary to data from the population-based German birth cohorts GINIplus (German Infant Nutritional Intervention plus environmental and genetic influences on allergy development) and LISAplus (influences of lifestyle-related factors on the immune system and the development of allergies in childhood plus air pollution and genetics) indicating that maternal BMI and overweight status were correlated with children’s intake of meat and egg products [39], we observed that children with at least one or both overweight/obese parents showed a lower percentage of total protein compared to those with no overweight/obese parents. Moreover, while the literature demonstrates a shorter duration of breastfeeding among overweight and obese women compared to mothers with a healthy BMI [40,41], we did not find any differences.

We found that the number of offspring was correlated with an earlier introduction of cow’s milk and a higher weekly frequency of SSB consumption before 24 months, suggesting a higher intake of protein and sugars at the expense of fats. Evidence from the Australian Healthy Smiles Healthy Kids birth cohort study including children from birth to the age of 3 years suggested that compared to low SSB consumers, high SSB consumers were significantly more likely to be living in households with three or more children [42]. It is possible that in families with a greater number of members, child care is more challenging, making it difficult for parents to adopt adequate feeding practices. Furthermore, a positive correlation has been found between maternal BMI and paternal BMI. Indeed, both obesity status and BMI changes among spouses were found to be associated over time, suggesting an impact of the home environment on modeling eating behaviors and nutritional status, that is, the sharing of similar obesogenic lifestyles such as unhealthy diets [43].

### Limits and Strengths

To the best of our knowledge, this is the only Italian study which explores feeding practices, eating behaviors, and nutrient intakes among Italian preschoolers in relation to their parents’ nutritional status, that is, parents being of a healthy weight or overweight/obese. Indeed, globally, there is a general paucity of studies dealing with this issue [6,7,8,9,39]. Thus, the current findings may contribute to filling such a gap in the knowledge and, possibly, to the suggesting of personalized interventions. A further strength lies in this study taking into consideration several feeding practices and child eating behaviors, and a broad spectrum of risk factors, thus providing a holistic view of the issue. However, there are some limitations. Firstly, due to the cross-sectional design, causal relationships cannot be inferred. Secondly, survey data were gathered from parents, which may have introduced recall bias. Thirdly, the accuracy of the information regarding children’s food consumption could also be compromised, as their meals did not always occur under parents’ supervision (i.e., school, other caregivers). Lastly, the results cannot be generalized, with the sample being representative of a narrow segment of the population. In particular, socio-demographic factors should be more precisely investigated, such as income, education, employment status, the type of job, the neighborhood of residence, etc.

## 5. Conclusions

Nutrition and food choices during early life may contribute to health and disease risk in later life, mostly in the context of public health concerns such as obesity. In our population, the percentage of children being exclusively breastfed was relevant, and the age at the introduction of complementary feeding met the recommendations. In contrast, the consumption of fruit juice as well as the intake of protein and sugars surpassed the national and international guidelines. Our findings underline that reducing and delaying the intake of SSBs should be a priority to promote a healthy weight through early childhood and growth. Moreover, they suggest that families, mostly the largest ones, should be the key for target-nutritional education, highlighting the importance of replacing sugary drinks with healthier and more nutrient-rich alternatives (i.e., whole-fruit or home-made fresh juices without any additives or refined sugars) across the life course to encourage the development of healthy eating patterns in offspring, thus preventing overweight and obesity. In this regard, by promoting healthy choices over industrial products, scientists and health workers may play a crucial role in counseling parents/caregivers. Future research is needed to better understand the associations between childhood eating behaviors and parental factors, including analyzing their socio-demographic characteristics. 

## Figures and Tables

**Figure 1 nutrients-17-00575-f001:**
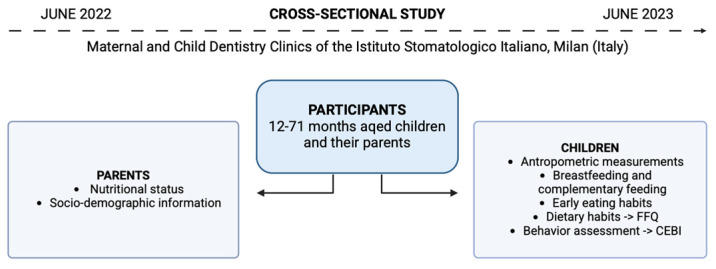
Study design’s main characteristics.

**Table 1 nutrients-17-00575-t001:** General and anthropometric characteristics of children (n = 171).

Characteristics		No	(%)
Age (years), mean (±SD)			4.6 (±1.1)
Male		100	58
Female		71	42
Ethnicity			
	Caucasian	158	92
	Hispanic	8	5
	Asian	4	2
	African	1	1
Birth weight < 2500 g		10	5.8
Born preterm		5	2.9
Weight (kg), mean (±SD)			17.6 (±3.0)
Length/height (m), mean (±SD)			1.1 (±0.1)
Nutritional status			
	Healthy weight	134	78
	Overweight	12	7
	Obese	1	1
	Underweight	24	14

**Table 2 nutrients-17-00575-t002:** Children’s feeding practices in the study population (n = 171).

Feeding Practices		Mean (±SD)
Feeding practices		mean (±SD)
Duration of EB (months)		5.9 (±6.8)
Age at introduction (months)		
	Infant formula	3.2 (±3.0)
	Growing-up milk	13.5 (±5.9)
	Cow milk	16.4 (±9.1)
	Complementary food	6.2 (±1.7)
Self-eating (months)		16.5 (±8.6)
Age at introduction (years)		
	SSBs	2.2 (±1.3)
Frequency of SSBs consumption before 24 months (times/week)		3.6 (± 4.3)
Frequency of consumption (times/week)		
	Fruit juice	3.2 (±2.9)
	Herbal tea	0.4 (±1.2)
	Tea	0.6 (±1.8)
	Chamomile tea	0.4 (±1.3)
Meals (number/day)		4.8 (±0.6)

EB: exclusive breastfeeding; SSBs: sugar-sweetened beverages.

**Table 3 nutrients-17-00575-t003:** Energy and nutrient intakes of children (n = 167).

Dietary Intake	Mean ± SD
Energy (kcal)	1375 ± 238
Proteins (g/kg)	3.0 ± 0.6
Proteins (%)	15 ± 2
Fats (g)	45 ± 13
Fats (%)	30 ± 7
Saturated fats (%)	11 ± 3
Carbohydrates (g)	201 ± 41
Carbohydrates (%)	55 ± 6
Sugars (%)	20 ± 8
Ca (mg)	566 ± 172
P (mg)	827 ± 186
Fe (mg)	7 ± 2

**Table 4 nutrients-17-00575-t004:** General and anthropometric characteristics of parents.

Characteristics		Mother (n = 171)	Father (n = 168)
Age (mean ± SD)		38.2 (±5.8)	41.3 (±7.2)
Educational level (No, %)			
	Master’s degree/PhD	87 (51)	59 (35)
	High school	62 (36)	80 (48)
	Secondary	22 (13)	29 (17)
Unemployed		29 (17)	2 (1)
Smoking			
	Currently	43 (25)	49 (29)
	During pregnancy	9 (5)	44 (26)
BMI (mean ± SD)		23.1 (±4.5)	25.7 (±3.5)
Nutritional status, n (%)			
	Healthy weight	111 (65)	81 (48)
	Underweight	17 (10)	1 (1)
	Overweight	30 (13)	65 (39)
	Obese	13 (8)	20 (12)

**Table 5 nutrients-17-00575-t005:** Nutritional status distribution of parents within families (n = 171), i.e., no obese/overweight parents (i.e., underweight or healthy weight) versus overweight/obese parents (i.e., at least one or both).

Parent Nutritional Status Distribution (No, %)	
No overweight or obese	71 (42)
1 overweight	59 (34)
1 obese	12 (7)
1 overweight and 1 obese	9 (5)
2 overweight	14 (8)
2 obese	6 (4)

**Table 6 nutrients-17-00575-t006:** Children’s feeding practices and dietary intake expressed as mean (± SD) based on nutritional status of parents, namely, not obese/overweight (i.e., underweight or healthy weight) versus obese/overweight (i.e., at least one or both).

		Parental Nutritional Status	
		No obese/overweight (n = 71)	Obese/overweight (n = 100)
Children’s feeding practices			
Duration of EB (months)		5.5 (±1.4)	5.3 (±1.9)
Age at introduction (months)			
	Infant formula	3.7 (±2.6)	3.2 (±2.5)
	Growing-up milk	11.6 (±2.9)	14.9 (±7.0) *
	Cow milk	14.9 (±6.1)	17.5 (±10.7)
	Complementary food	6.2 (±1.3)	6.2 (±2.0)
Self-eating (months)		15.8 (±7.3)	17.0 (±9.4)
Frequency of SSBs consumption (times/week) before 24 months		3.4 ± 2.9	3.8 ± 5.0
Dietary intake			
	Energy (kcal)	1332 ± 276	1393 ± 230
	Proteins (g/kg)	3.0 ± 0.7	2.9 ± 0.6
	Proteins (%)	16 ± 2	15 ± 2 **
	Fats (g)	16 ± 5	16 ± 5
	Fats (%)	29 ± 7	30 ± 7
	Saturated fats (%)	10 ± 3	11 ± 2
	Carbohydrates (g)	196 ± 43	205 ± 39
	Carbohydrates (%)	54 ± 9	55 ± 6
	Sugars (%)	19 ± 8	20 ± 7
	Ca (mg)	567 ± 198	559 ± 154
	P (mg)	816 ± 192	828 ± 191
	Fe (mg)	7 ± 2	7 ± 1

EB: exclusive breastfeeding; SSBs: sugar-sweetened beverages. Student’s *t*-test: ** *p* < 0.01; * *p* < 0.05.

**Table 7 nutrients-17-00575-t007:** Children’s feeding practices and dietary intake expressed as mean (± SD) based on nutritional status of parents, namely, both obese or overweight parents versus the other parents (i.e., not obese/overweight or only one obese/overweight).

		Parental Nutritional Status	
		Other parents (n = 141)	Both obese or overweight (n = 30)
Children’s feeding practices			
Duration of EB (months)		5.3 (±1.6)	5.7 (±2.0)
Age at introduction (months)			
	Infant formula	3.2 (±2.4)	3.9 (±3.0)
	Growing-up milk	13.0 (±5.2)	15.2 (±7.8
	Cow milk	16.3 (±9.3)	16.9 (±8.4)
	Complementary food	6.1 (±1.1)	6.4 (±3.4)
Self-eating (months)		16.0 (±8.3)	18.7 (±9.7)
Frequency of SSBs consumption (times/week) before 24 months		3.4 (±3.3)	4.5 (±7.2)
Dietary intake			
	Energy (kcal)	1354 ± 253	1433 ± 240
	Proteins (g/kg)	3.0 ± 0.6	3.0 ± 0.6
	Proteins (%)	15 ± 2	15 ± 2
	Fats (g)	45 ± 13	47 ± 15
	Fats (%)	30 ± 7	29 ± 6
	Saturated fats (%)	11 ± 3	10 ± 2
	Carbohydrates (g)	199 ± 41	212 ± 37 *
	Carbohydrates (%)	54 ± 8	56 ± 5
	Sugars (%)	20 ± 8	20 ± 4
	Ca (mg)	565 ± 178	553 ± 152
	P (mg)	824 ± 201	815 ± 140
	Fe (mg)	7 ± 2	7 ± 1

EB: exclusive breastfeeding; SSBs: sugar-sweetened beverages. Student’s *t*-test: * *p* < 0.05.

## Data Availability

The data that support the findings of this study are contained within the article.

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
