# Peer review of "Preschool Children’s Eating Habits and Parental Nutritional Status"

_nutrients, 2025, doi:10.3390/nu17030575_

Round 1

Reviewer 1 Report

Comments and Suggestions for Authors

The manuscript by Silvia Bettocchi et al. aimed to characterize eating habits among preschoolers and investigate the relationship with parents’ nutritional status. The study is well-designed and the findings are interesting. I have the following questions and comments:

1, line 131 to 133 should be deleted. 

2, the authors claimed that this is the only Italian study conducted. However, the authors should discuss the findings in a broad background. What about the findings in other countries? This must be discussed. 

3, a figure illustrating the design of the study should be added. 

4, the authors suggested that  family-targeted interventions must be implemented. However, the authors have not discussed about the interventions in the manuscript. Could the authors give more thoughts about this? What are the interventions?

Author Response

Response to Reviewer 1 Comments

1. Summary

Thank you very much for taking the time to review this manuscript. We would like to thank the Reviewer for the important comments and suggestions which have contributed to improving the manuscript’s quality.

2. Questions for General Evaluation

Reviewer’s Evaluation

Does the introduction provide sufficient background and include all relevant references?

Yes

Are all the cited references relevant to the research?

Yes

Is the research design appropriate?

Yes

Are the methods adequately described?

Yes

Are the results clearly presented?

Yes

Are the conclusions supported by the results?

Yes

3. Point-by-point response to Comments and Suggestions for Authors

Comments 1: line 131 to 133 should be deleted.

Response 1: Thank you for pointing this out. We agree with this comment. We have deleted

Comments 2: the authors claimed that this is the only Italian study conducted. However, the authors should discuss the findings in a broad background. What about the findings in other countries? This must be discussed.

Response 2: Agree. We have, accordingly, revised the main text to emphasize this point. Globally, there is a general paucity of research dealing with childhood eating behaviors based on the parents’ nutritional status in the first years of age. In the previous version of the manuscript, we cited the above-mentioned research in the Introduction (references 6-9) and Discussion (reference 39) without referring to the Countries where the studies were carried out. In the current version, we provide such details in the Introduction (page 2) and the references to the above-mentioned studies in Limits and Strengths (page 11), as suggested by the Reviewer to provide a broader background to our findings.

Comments 3: a figure illustrating the design of the study should be added.

Response 3: Agree. We have, accordingly, added Figure 1 in a separate file as reported in the main text (line 89).

Comments 4: the authors suggested that  family-targeted interventions must be implemented. However, the authors have not discussed about the interventions in the manuscript. Could the authors give more thoughts about this? What are the interventions?

Response 4: The Reviewer is right. We meant that as the consumption of fruit juice and the intake of sugars in our sample overcame the national and international guidelines there is the need to counsel/educate families. Therefore, we have replacedfamily-targeted interventions must be implemented” with “nutritional education aimed at families is needed” in the Abstract. Furthermore, we have implemented this issue in Conclusions “… family, mostly the largest ones, should be the key for target-nutritional education highlighting the importance of replacing sugary drinks with healthier and more nutrient-rich alternatives (i.e., whole fruit or home-made fresh juices without any additives or refined sugars) across the life course to encourage the development of healthy eating patterns in offspring, thus preventing overweight and obesity. In this regard, by promoting healthy choices over industrial products, scientists and health workers may play a crucial role in counseling parents/caregivers…”.

4. Response to Comments on the Quality of English Language

Point 1: The quality of English does not limit my understanding of the research

Response 1: We would like to thank the Reviewer for this comment.

Reviewer 2 Report

Comments and Suggestions for Authors

very interesting and valuable article regardless of the study limitations that are stated
a couple of remarks: I would certainly at least briefly describe complementary food, it is mostly according to the pediatrician's recommendation, but even there can be hidden traps of calorie intake.
also perhaps one should comment on the intake of fresh fruit and juices that may be given by parents made from fresh fruit
and at the end of the text it is noted that sweetened juices should not be given to children under a certain age (2 years), however, scientific research indicates how dangerous refined sugar is for the human population as a whole and how much we are exposed to sugar during our lives, and I am of the opinion that the later children are exposed to sugar the better. a healthier alternative is always fresh juices without any additives, particularly sugars. we all know this is very hard to achieve, but still, we are scientists and some of us health workers, and we should always promote healthy choices over industrial products.

Author Response

Response to Reviewer 2 Comments

1. Summary

Thank you very much for taking the time to review this manuscript. We would like to thank the Reviewer for the important comments and suggestions which have contributed to improving the manuscript’s quality.

2. Questions for General Evaluation

Reviewer’s Evaluation

Does the introduction provide sufficient background and include all relevant references?

Yes

Are all the cited references relevant to the research?

Yes

Is the research design appropriate?

Yes

Are the methods adequately described?

Yes

Are the results clearly presented?

Yes

Are the conclusions supported by the results?

Yes

3. Point-by-point response to Comments and Suggestions for Authors

Comments 1: very interesting and valuable article regardless of the study limitations that are stated

a couple of remarks: I would certainly at least briefly describe complementary food, it is mostly according to the pediatrician's recommendation, but even there can be hidden traps of calorie intake.

also perhaps one should comment on the intake of fresh fruit and juices that may be given by parents made from fresh fruit

and at the end of the text it is noted that sweetened juices should not be given to children under a certain age (2 years), however, scientific research indicates how dangerous refined sugar is for the human population as a whole and how much we are exposed to sugar during our lives, and I am of the opinion that the later children are exposed to sugar the better. a healthier alternative is always fresh juices without any additives, particularly sugars. we all know this is very hard to achieve, but still, we are scientists and some of us health workers, and we should always promote healthy choices over industrial products.

Response 1: We would like to thank the Reviewer for the important comments and suggestions which have contributed to improving the manuscript’s quality.

Comments 2: Regarding complementary feeding, the reviewer is right: The nutritional quality is important too. In the current version, we describe briefly the topic. Even though in our study we asked only for the age of introduction of complementary food, we also collected the age of introduction of some beverages that may provide some (even if partially) information about the quality of complementary feeding. We discuss this in several parts of the manuscript (pages 9, 10, 11, 12). 

Regarding sugars, we fully agree with the Reviewer, we just reported the current national and international recommendations for children. In the present version, we have implemented the issue according to the Reviewer’s suggestions (pages 10 and 11). 

Point 1: The quality of English does not limit my understanding of the research

Response 1: We would like to thank the Reviewer for this comment.

Round 2

Reviewer 1 Report

Comments and Suggestions for Authors

Figure 1 is missing in the manuscript. Please revise. 

Author Response

Figure 1  has been added.